# Comparison of Neural Networks and Gradient Boosting Models on Ordinal Age Class Prediction using Mouse Trajectories

Tobias Wistuba[1], Lisa Bondo Andersen[2,3], Ailin Liu[2,3], Felix Henninger[2,3], Frauke Kreuter[2,3], and Sonja Greven[1]

[1]Humboldt University, Berlin
[2]Ludwig Maximilian University of Munich
[3]Munich Center for Machine Learning

## Abstract

We aim to predict ordinal age classes via mouse trajectories collected through web surveys. We compare performance of different neural network architectures, including dense neural networks using the entire trajectory as input, 1D- and 2D-convolutional neural networks, long short-term memory neural networks, and transformers against gradient boosting models that use hand-crafted features of the trajectories as inputs. The results show that neural networks as well as gradient boosting models are able to predict age classes with accuracies above pure chance. However, despite their higher complexity, neural networks do not clearly outperform the boosting models and do not offer the advantages of supplying interpretable results and detecting informative covariates.

## 1 Introduction

Simple paradata like response time is easy to capture and informative for tasks such as cognitive impairment detection [1]. However, richer paradata such as cursor trajectories captured during computer tasks provide a more nuanced view of participant behavior, and might further improve predictions. For example, Fernández-Fontelo et al. [2] were able to predict the difficulty respondents faced using manually defined mouse trajectory features collected in online surveys. Due to the complexity and volume of the data, neural networks promise to further improve prediction based on trajectory data. Building on previous work showing a relationship between age and computer mouse movements [3, 4], we predict age via mouse trajectories, comparing neural networks and gradient boosting models. Ultimately, we want to determine whether it is beneficial to use more complex neural networks over classical machine learning approaches.

## 2 Methods

The neural network architectures we consider are: dense neural networks (DNN), 1D-convolutional

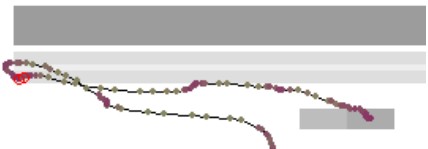

**Figure 1.** Example trajectory of a survey question as input for the 2D-convolutional neural networks. Clicks are shown as red circles.

neural networks (1D-CNN), 2D-convolutional networks (2D-CNN), Long Short-Term Memory networks (LSTM), and transformers. We chose DNNs as a baseline comparison for the other architectures, 1D-CNNs and LSTMs due to their ability to handle time series data, 2D-CNNs to work with image visualizations of the trajectories, and transformers because of their current success and popularity.

### 2.1 Preprocessing

The form of the input differs between the selected architectures and therefore so does the related preprocessing of the data. We firstly preprocessed all raw trajectories using the `mousetrap` package [5]. All raw trajectories were time-normalized to achieve a uniform sequence length of 101 sampling points. We then added temporal features, i.e. whether a click happened, time between different timestamps, acceleration, and velocity of the mouse at a given time point $t$, since they were informative in previous research [2, 6]. Finally, we included layout information such as the positions of question and answer text as inputs to provide context to the trajectories. This input was concatenated to the final layers of each neural network, with exception of the 2D-CNNs, after handling of the trajectory covariates. The layout was directly processed by the 2D-CNNs alongside the trajectories (see figure 1). We modeled ordinal responses within the neural networks via the CORAL framework [7]. As a second model class, we chose **Gradient Boosting** models as they showed promising results in the work of Fernández-Fontelo et al. [8] handling similar data. They were also straightforward to implement via `xgboost` and the

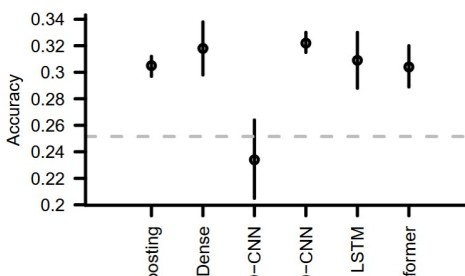

**Figure 2.** Accuracies of different modeling approaches. Circles show the mean value over all 10 folds and vertical lines the standard deviation over all 10 folds. The dotted line shows the accuracy when just predicting the most prevalent age class.

|  | Pred 18-30 | Pred 31-42 | Pred 43-54 | Pred 55-66 | Pred 67-78 | Pred 79+ | True |
|---|---|---|---|---|---|---|---|
| True 18-30 | 78 | 182 | 113 | 60 | 12 | 3 | 448 |
| True 31-42 | 85 | 214 | 205 | 116 | 22 | 6 | 648 |
| True 43-54 | 53 | 261 | 277 | 255 | 147 | 23 | 1016 |
| True 55-66 | 11 | 117 | 270 | 436 | 366 | 56 | 1256 |
| True 67-78 | 3 | 41 | 123 | 288 | 398 | 136 | 989 |
| True 79+ | 3 | 8 | 20 | 80 | 159 | 82 | 352 |
| Pred | 233 | 823 | 1008 | 1235 | 1104 | 306 | 0.315 |

**Figure 3.** Confusion matrix on a test dataset of a transformer model. The accuracy is displayed in the bottom right corner. Brighter colors indicate higher prediction counts.

mlr3 package [9].

For **DNNs**, the seven time-normalized functional variables (timestamps, x-positions, y-positions, clicks, distance, velocity, and acceleration) were stacked to create an input vector $\vec{x} \in \mathbb{R}^{707}$. For **1D-CNNs and Transformers**, the seven time-normalized functional variables were stacked in such a way as to create an input matrix $\mathbf{X} \in \mathbb{R}^{101 \times 7}$. For the **2D-CNNs**, visualizations of the trajectories as 300x300 pixel images were used as inputs. Similarly to Niu et al. [6], we encoded acceleration and velocity at a given point in the red, green, and blue color channels (see the example trajectory in figure 1).

## 2.2 Data

Our data was collected in a wave of the Understanding America Study, a survey panel with 14,000 participants. Trajectories were filtered to remove touchscreen trajectories. The total sample size was 26520 trajectories from 3315 respondents answering 8 questions. We assessed model performance through 10-fold-cross-validation. The validation dataset was used for early stopping. All models were trained and evaluated using the same folds, created such that all the trajectories of one participant were only part of either the training dataset, the validation dataset, or the testing dataset within each fold. The age classes and counts are 18-30: 2280, 31-42: 4144, 43-54: 5056, 55-66: 6672, 67-78: 6592, 79+:1776.

## 3 Results

Since the response variable is unbalanced, we compare the performances of the different approaches to the majority class accuracy of 25.2%. The mean accuracies (±1 standard deviation) on the testing data can be seen in figure 2. We also show an example of a confusion matrix in figure 3.

## 4 Discussion and Conclusion

In this work, we show the usefulness of mouse trajectories as a predictor for age class classification. As a representative result, a trend towards the diagonal of the confusion matrix can be seen in figure 3. The poor performance of 1D-CNNs might be an indicator of incorrect model architecture. Choosing a different one could possibly improve the performance. However, even though neural networks offer higher complexity, the gradient boosting method was able to compete with their accuracies and did not perform substantially worse. In the case of age class prediction, opting for more complex neural networks does not result in clearly better performances and machine learning models on hand-crafted features appear to be a valid approach. Additionally, users benefit from interpretability of results and determination of informative features. Out of the tested network architectures, 2D-CNNs performed best, but with only small improvements over boosting. It is still possible that a specialized neural network architecture might perform better on this type of data and improve prediction accuracies. In conclusion, our results - just like the works of Fernández-Fontelo et al. [2] and Thorpe et al. [10] - show that paradata is informative and can possibly be used for different prediction tasks.

## Acknowledgments

We would like to thank our collaboration partners as well as the participants of the Understanding America Study for their valuable contributions. Funded by the Deutsche Forschungsgemeinschaft (DFG, German Research Foundation) – project number 396057129 ("Statistical modeling using mouse movements to model measurement error and improve data quality in web surveys").

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
