# OpenReview forum: "Comparison of Neural Networks and Gradient Boosting Models on Ordinal Age Class Prediction using Mouse Trajectories"
_NLDL.org/2026/Abstracts_Track — NLDL 2026 Abstracts_

### Official Review · Reviewer_9gFm · 2025-10-31

**Soundness:** 3
**Correctness:** 3
**Rating:** 4
**Confidence:** 5

**Summary:**

The authors propose a pipeline to predict age from mouse trajectories, which could be a marker for cognitive loss or impairment.
They compare different neural networks (CNNs, transformers) to standard machine learning techniques like gradient boosting.

**Strengths:**

The abstract is well written and clear.
The figures support the textual claims.
The experimental setup is sound and correct.
The results can be interesting for the NLDL community, in particular because this is not a very usual task so it can spark some interesting discussion.

**Weaknesses:**

No significant weaknesses.

---

### Official Review · Reviewer_giBW · 2025-10-31

**Soundness:** 3
**Correctness:** 3
**Rating:** 4
**Confidence:** 2

**Summary:**

The abstract is about trying to predict the age category of given individual from mouse trajectory on a web survey. The authors compared multiple neural architectures (Dense, CNN, LSTM, Transformers) against Gradient boosting with handcrafted features, obtaining similar results. Confusion matrix indicates, that the trajectory indeed captures some age based data as clear diagonal line can be observer, even if the overall accuracy is relatively low.

**Strengths:**

Well written with great presentation and the inclusion of multiple architectures.
The provided confusion matrix and confidence ranges is really helpfull in assesing the performence.

**Weaknesses:**

Not clear how the layout information was integrated into the input vectors as they seem to be just the 101 points with the time-normalised functional variables (timestamps, x-positions, y-positions, 077 clicks, distance, velocity, and acceleration)

Visual Transformer could process series of images and obtain the information by layer stacking as done in `Playing Atari with Deep Reinforcement Learning`. This was not explored.

Data description is little bit confusing. First we hear of 14 000 participants, but lower we get 3 315 Participants and 26 520 Trajectories. The amount of 14 000 is therefore irrelevant.

While the authors asses the baseline as 25.2 %, this baseline is different across each test fold. For example for the test fold present in Figure 3, the baseline is 26.7 %

---

### Official Review · Reviewer_jPER · 2025-11-02

**Soundness:** 4
**Correctness:** 4
**Rating:** 5
**Confidence:** 3

**Summary:**

The authors seek to predict/classify age groups based on mouse trajectories and clicks. They try a variety of models including DNNs, Transformers, LSTMs, CNNs (1D, 2D), and gradient boosting methods.
They pre-processed their temporal location data along with hand crafted features (clicks, velocity, acceleration, time diff) such that the different model types would work smoothly with the inputs.
They train their models with 10-fold cross-validation on the roughly 27k samples. They show similar performance for most models, with the exception of the 1D CNN being much worse, and the 2D CNN being slightly better and more consistent than the other options but still within error bounds. Notably, the 2D CNN also had the most involved preprocessing that essentially reconstructed a "video" of the movement as a single image frame.
They conclude similar trends as previous work and note that the success of the gradient boosting method could be preferably, given its classical foundations and simplicity when compared to the modern neural networks.

**Strengths:**

- Nice with error bounds from the 10-fold cross-validation
- Good and varied selection of model types
- Generally well-written and clear in its language.
- Good that the preprocessing is explained in detail for the model types, since it can be very important and is often glossed over.

**Weaknesses:**

- There is no specification of how many parameters the different networks contained, and considering this could have an significant influence on performance it seems necessary to account for. Were networks sizes normalized to be similar?
- With many models achieving a similar performance, it would be interesting to see a class-balanced accuracy plot instead of standard accuracy,, or a similar metric that takes into account that the sizes of the age groups are different.
- Was the fact that the classes are ordinal used anywhere? Isn't calling age groups "ordinal age groups" a bit redundant, I can't think of any non-ordinal age groups.

---

### Decision · Program_Chairs · 2025-11-05

**Decision:**

Accept

**Comment:**

The abstract is of interest to the community and should be presented at the conference.